# Less Is More for Non-Dislocated Femoral Neck Fractures: Similar Results for Two versus Three Cannulated Hip Screws

Hilde Schutte *, Lorenzo Hulshof, Ger van Olden, Paul van Koperen, Tim Timmers and Wouter Kluijfhout

Department of Trauma Surgery, Meander Medical Center, 3813 TZ Amersfoort, The Netherlands
* Correspondence: hilde.n.schutte@gmail.com

**Abstract:** Cannulated hip screws (CHS) can be used for the minimally invasive fixation of non-dislocated femoral neck fractures. Usually, three screws are inserted. This study aims to determine whether fixation by two CHS leads to similar results as fixation by three CHS. Since January 2019, all patients with an indication for internal fixation by CHS were treated with two CHS and followed prospectively. Results were compared to an equal-sized control group of patients who underwent fixation by three CHS (before 2019). The primary outcome was reoperation, while the secondary outcome was screw dislocation. Since January 2019, 50 patients were treated by two CHS. Of these, 14 patients (28%) underwent reoperation versus 13 patients (26%) in the control group ($p = 1.000$). Reoperations included screw replacement, hemiarthroplasty, and total hip prosthesis. Three major reasons for reoperation were pain due to osteosynthesis material ($n = 15$), coxarthrosis ($n = 4$), and screw cut out ($n = 3$). Six weeks postoperative X-rays showed a screw dislocation of 2 mm for the two CHS group and 1 mm for the three CHS group ($p = 0.330$). Clinical outcomes were very similar between the groups. The overall results were good; however, the reoperation rate varied from 26 to 28%. The majority of reoperations were screw replacements. Screw dislocation seems to be more prominent in patients treated with two screws (2 mm versus 1 mm). Fixation by two cannulated hip screws is an acceptable treatment method for non-dislocated femoral neck fractures, and the insertion of a third screw does not lead to superior clinical results.

**Keywords:** hip fractures; femoral neck fractures; fracture fixation; internal; postoperative complications; reoperation

## 1. Introduction

Hip fractures represent a significant global health issue, with an estimated incidence of 1.26 million cases worldwide in 1990. It is projected that by the year 2050, this number will quadruple [1]. Apart from the high morbidity and mortality [2,3], hip fractures contribute substantially health care costs. Per-patient costs amount to EUR 27.500 in the first year alone [4].

Numerous risk factors contribute to the incidence of femoral neck fractures, including female gender, low bone density, and hypertension [5–7].

Over the years, researchers have sought to obtain insight into the optimal treatment method for proximal femur fractures. Surgical treatment options include internal fixation, hemiarthroplasty, and total hip arthroplasty [8]. The optimal surgical treatment method depends on case specific factors including fracture type, age, and comorbidity of the patient [9], and conservative treatment might also be beneficial in some cases [10].

Choice of treatment method relies predominantly on fracture type. The Garden classification is the most commonly utilized system to classify femoral neck fractures [11]. The system classifies femoral neck fractures based on the degree of displacement and fracture pattern, as seen on anterior to posterior (AP) radiographs. Garden Class I and II fractures are often considered stable and may be amenable to internal fixation techniques. In contrast, Garden Class III and IV fractures are typically unstable and require more

extensive surgical interventions such as arthroplasty or internal fixation with additional adjuncts to achieve adequate stability [12].

For non-dislocated femoral neck fractures, current guidelines indicate internal fixation through cannulated hip screws (CHS). Screws are inserted following the "three-point principle", by inserting three cannulated screws in an inverted triangular pattern [13–17] (Figure 1). This method shows excellent results in the management of Garden I and II femoral neck fractures [18]. However, the question arises as to whether it is necessary to insert three screws, or if two screws can yield comparable clinical results.

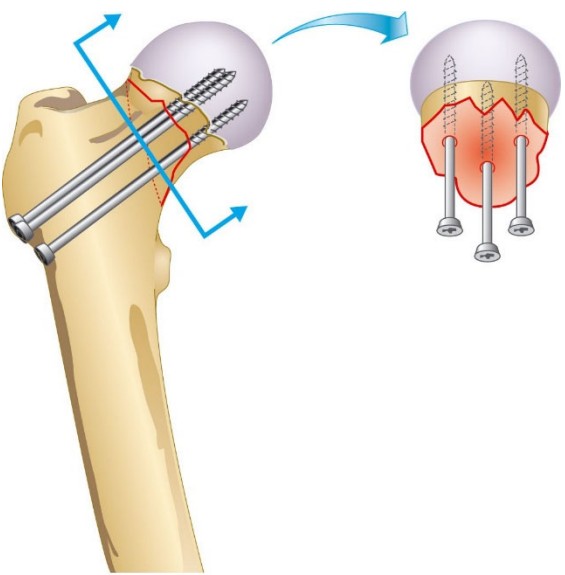

**Figure 1.** Triangular insertion pattern of three cannulated hip screws.

In biomechanical studies, three CHS fixation can bear significantly higher weight than fixation with two CHS [19–21]. Yet, the clinical relevance of this increased weight-bearing capacity remains uncertain. In cases where the femoral neck could not accommodate three CHS due to its diameter, the insertion of two CHS has been justified [22]. Previous studies have shown that two screws provide adequate fixation [23–25]. Fixation with two screws offers several advantages, such as reduced surgery time, decreased operative irradiation, lower costs, and a less technically demanding procedure [26]. Moreover, radiological evidence suggests that union occurs more frequently in two-screw fixation than in three-screw fixation [27]. Despite these findings, there is a scarcity of literature on the clinical outcomes associated with fixation using two CHS. Therefore, the primary objective of this study is to investigate whether fixation of non-dislocated femoral neck fractures (Garden I and II) using two CHS yields similar results as fixation by three CHS. Screw displacement was considered as a secondary outcome parameter.

## 2. Materials and Methods

### 2.1. Study Design

This study was conducted at the Meander Medical Center, Amersfoort, The Netherlands. This study was approved by the local ethics committee and performed in accordance with the ethical standards (study number TWO 20-063). The duration of the study extended from 1 January 2019 to December 2020. Based on the literature, our trauma department decided to change their treatment strategy in January 2019 for patients with non-dislocated femoral neck fractures. Instead of inserting three cannulated screws, Garden 1 or 2 femoral neck fractures were fixated with two screws without patient selection. All patients entered a prospectively maintained database to carefully evaluate this change in practice. All patients who underwent fixation by two CHS until December 2020 were included in this study. Controls with three CHS were identified in consecutive order until a correspond-

ing number of patients was obtained. Patients with dislocated femoral neck fractures (Garden III and IV) were excluded, as well as patients who were lost to follow-up (who did not visit the outpatient clinic).

## 2.2. Surgical Technique

A dose of 2 g cefazolin is given between 30 and 60 min preoperatively as antimicrobial prophylaxis. Surgery is performed under a supine position, with application of traction and internal rotation of the hip. Before incision, reduction of the fracture is verified using X-ray guidance. A 3 cm incision is made, starting at the flair of the great trochanter and extending distally. The fascia lata is split, and the vastus lateralis is held aside to reveal the femoral cortex.

Kirschner (K-)wires are placed under X-ray guidance for optimal positioning of the screws in two directions. The first K-wire is placed inferiorly, over the calcar, right in the middle of the femoral cortex. The second K-wire is placed superior to the first and more posterior. The length of the K-wires is measured to determine screw lengths. The cannulated 6.5 mm screws are placed over the K-wires, starting with the posterior screw to prevent varus deformity. After placement of both screws, positioning is checked by X-ray in both directions. For the positioning of the screws in two CHS fixation, see Figure 2. The fascia lata is closed with an absorbable continuous suture. The skin is closed with a non-absorbable suture. Directly after operation, a physiotherapist is consulted to support patients in mobilizing. Postoperative prophylaxis for venous thromboembolism is carried out based on established protocols, taking into consideration the patient's current anticoagulant status. The administration of low-molecular-weight heparins (LMWH) is the prevailing method used for this purpose. Clinical rehabilitation usually takes five days, depending on the patient's age and comorbidities. Patients are instructed to mobilize at 50% weight bearing with crutches or a walker. If the clinical situation allows so, patients are discharged home. If not, patients are transferred to a rehabilitation facility. Sutures will be removed by the general practitioner, two weeks after surgery. Six weeks postoperatively, patients return to the outpatient clinic for radiological and clinical follow-up. Patients are instructed to contact the hospital if any adverse events happen.

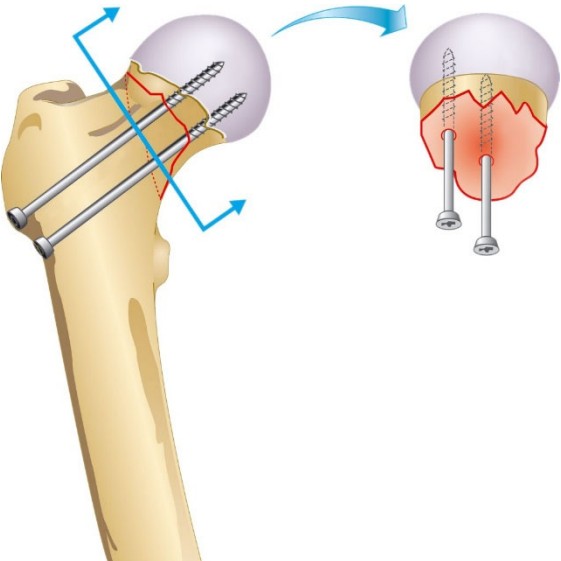

**Figure 2.** Insertion pattern of two cannulated hip screws.

## 2.3. Data Collection

Data on patients' demographics, fracture type, screw length, and screw translocation were obtained through electronic patient files. Renal function was categorized by the

2012 Kidney Disease Outcomes Quality Initiative (KDIGO) Chronic Kidney Disease (CKD) classification [28]. Fracture types were classified using the Garden Classification [12].

The intervention group included all patients who underwent fixation by two CHS between January 2019 and December 2020. The control group existed of an equal number of patients who underwent fixation by three CHS. The primary outcome parameter was reoperation and included screw replacement, conversion to hemiarthroplasty (HA), and conversion to total hip prosthesis (THP). A secondary outcome parameter was screw displacement. This was defined by lateral protrusion into soft tissue. Screw displacement was calculated six weeks postoperatively, using radiographic measurements in correlation with the actual screw lengths. For patients who underwent reoperation, screw displacement was measured once more, before reoperation. Follow-up duration was registered as full months up until the date of data extraction from the electronic patient's files. Patients who died before the six weeks outpatient check-up were excluded (loss to follow-up). For patients who died after the check-up, mortality was registered, and follow-up duration was calculated until the date of death.

### 2.4. Statistical Analysis

Statistical analysis was performed using the computing environment R (R Studio Team; 2021). Categorical data are shown as numbers with percentages, and significant differences were calculated through the chi-square test or Fisher exact test as appropriate. For continuous data, normality of distribution was assessed using quantile-quantile (Q-Q) plots. In the case of normal distribution, categorical data are shown as the mean with range. In the case of non-normal distribution, categorical data are shown as the median with interquartile range (IQR). Significant differences were calculated using Student's *t*-test or Mann–Whitney U-test, respectively. For all analyses, a two-sided *p*-value < 0.05 was used as the threshold of statistical significance.

## 3. Results

Between January 2019 and December 2020, a total of 53 patients underwent fixation of a femoral neck fracture with two CHS. Of these, three were lost to follow-up. A corresponding number of 50 control patients with non-dislocated femoral neck fractures who were treated with three CHS were identified in subsequent order. In total, 100 patients were included for analysis (Figure 3).

The average age was 71 years and ranged from 29 to 97 years. The majority of the sample (74%) was female. Distribution over American Society of Anesthesiologists (ASA) classes was as follows. Seventeen patients were ASA 1, 44 were ASA 2, 36 were ASA 3, and three were ASA 4. Kidney function was G1 for 29 patients, G2 for 52 patients, G3a for 10 patients, G3b for six patients, G4 for two patients, and G5 for one patient. The average time until patients visited the outpatient clinic was seven weeks. The majority (69%) suffered a Garden I type fracture, and the other 31% suffered a Garden II type fracture. Sixteen percent used a walking aid for mobilizing before the trauma. There were no significant differences in baseline characteristics between the intervention group and the control group (Table 1). Examples of postoperative radiographs are provided in Figures 4 and 5.

In total, 27 patients underwent reoperation after a median of six months. In the two and three CHS groups, reoperation rates did not differ (28% and 26% respectively, *p* = 1.000). Reasons for reoperations in the two CHS group were complaints of pain due to osteosynthesis material (OSM) in seven patients (50%), coxarthrosis in two (14%), screw cut-out in two (14%), avascular necrosis (AVN) in one (7%), screw dislocation in one (7%), and revision to HA due to a fall in one (7%).

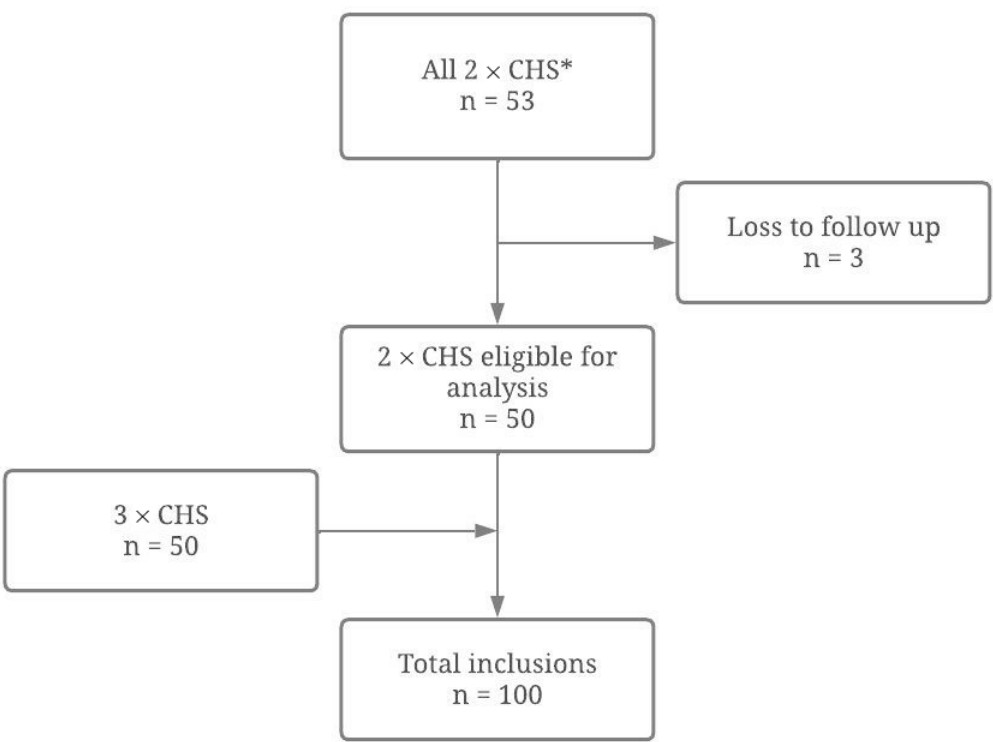

**Figure 3.** Flowchart depicting inclusion of study subjects. CHS: cannulated hip screws. * Between January 2019 and December 2020.

**Table 1.** Baseline characteristics [a].

| | | Overall (*n* = 100) | 2 CHS (*n* = 50) | 3 CHS (*n* = 50) | *p* |
|---|---|---|---|---|---|
| Age, mean (range) | | 71 (29–97) | 72 (29–91) | 70 (38–97) | 0.427 |
| Female, *n* (%) | | 74 (74) | 35 (70) | 39 (78) | 0.494 |
| ASA-classification, *n* (%) | 1 | 17 (17) | 6 (12) | 11 (22) | 0.299 |
| | 2 | 44 (44) | 21 (42) | 23 (46) | |
| | 3 | 36 (36) | 22 (44) | 14 (28) | |
| | 4 | 3 (3) | 1 (2) | 2 (4) | |
| Kidney function class, *n* (%) | G1 | 29 (29) | 13 (26) | 16 (32) | 0.326 |
| | G2 | 52 (52) | 27 (54) | 25 (50) | |
| | G3a | 10 (10) | 7 (14) | 3 (6) | |
| | G3b | 6 (6) | 1 (2) | 5 (10) | |
| | G4 | 2 (2) | 1 (2) | 1 (2) | |
| | G5 | 0 | 0 | 0 | |
| Visit outpatient clinic [a], mean (SD) | | 7 (2) | 6 (2) | 7 (1) | 0.452 |
| Garden, *n* (%) | I | 69 (69) | 36 (72) | 33 (66) | 0.665 |
| | II | 31 (31) | 14 (28) | 17 (34) | |
| Walking aid, *n* (%) | Unknown | 5 (5) | 5 (10) | 0 | 0.070 |
| | Yes | 16 (16) | 8 (16) | 8 (16) | |
| | No | 79 (79) | 37 (74) | 42 (84) | |

Abbreviations: CHS, cannulated hip screw; SD, standard deviation; ASA, American Society of Anesthesiologists.
[a]: in weeks.

For the three CHS group, reasons for reoperation were complaints of pain due to OSM in eight patients (62%), screw cut-out in one (8%), coxarthrosis in two (15%), infection in one (8%) and non-union in one (8%, Table 2). This was the only patient from the total cohort who suffered a non-union. In all other patients, radiological union was confirmed after six weeks. A detailed description of each reoperation is provided in Table 3. This table demonstrates that for some patients, dislocation of screws progressed over time.

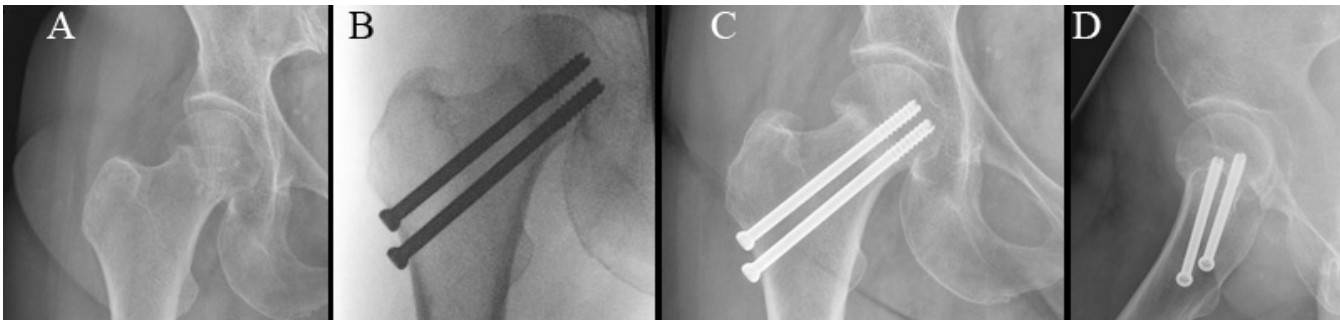

**Figure 4.** Radiographs of cannulated hip screw fixation with two screws. (**A**) Preoperative radiograph. (**B**) Perioperative radiograph. (**C**) Anteroposterior radiograph at 6-week follow-up. (**D**) Lateral radiograph at 6-week follow-up.

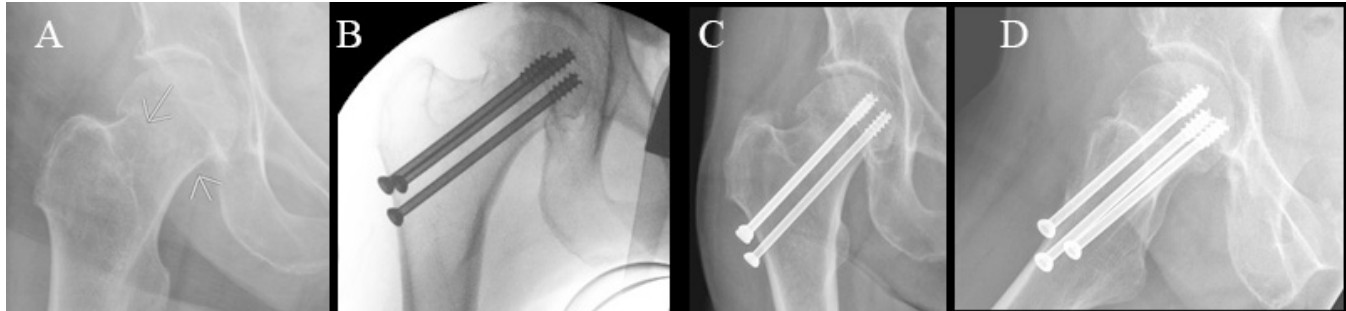

**Figure 5.** Radiographs of cannulated hip screw fixation with three screws. (**A**) Preoperative radiograph. (**B**) Perioperative radiograph. (**C**) Anteroposterior radiograph at 6-week follow-up. (**D**) Lateral radiograph at 6-week follow-up.

**Table 2.** Treatment methods and outcomes [a].

|  |  | Overall (*n* = 100) | 2 CHS (*n* = 50) | 3 CHS (*n* = 50) | *p* |
|---|---|---|---|---|---|
| Reoperation, *n* (%) |  | 27 (27) | 14 (28) | 13 (26) | 1.000 |
| Reoperation reason, *n* (%) | Pain OSM | 15 (56) | 7 (50) | 8 (62) | 0.615 |
|  | Coxarthrosis | 4 (15) | 2 (14) | 2 (15) |  |
|  | Screw cut out | 3 (11) | 2 (14) | 1 (8) |  |
|  | AVN | 1 (4) | 1 (7) | 0 |  |
|  | Screw dislocation | 1 (4) | 1 (7) | 0 |  |
|  | Fall | 1 (4) | 1 (7) | 0 |  |
|  | Infection | 1 (4) | 0 | 1 (8) |  |
|  | Non-union | 1 (4) | 0 | 1 (8) |  |
| Reoperation type [b], *n* (%) | Screw replacement | 15 (56) | 7 (50) | 8 (62) | 0.042 * |
|  | HA | 5 (19) | 5 (36) | 0 |  |
|  | THP | 7 (26) | 2 (14) | 5 (39) |  |
| Time to reoperation [c], median [IQR] |  | 6 [3–10] | 4 [2–7] | 7 [5–11] | 0.157 |
| Lateral protrusion [d], median [IQR] |  | 1 [0–5] | 2 [0–5] | 1 [0–4] | 0.330 |
| Mortality, *n* (%) |  | 7 (7) | 3 (6) | 4 (8) | 1.000 |
| Follow up [c], mean (SD) |  | 35 (15) | 26 (9) | 45 (12) | <0.001 * |

Abbreviations: CHS, cannulated hip screw; OSM, osteosynthesis material; AVN, avascular necrosis; HA, hemiarthroplasty; THP, total hip prosthesis; IQR, interquartile range; SD, standard deviation. [a]: due to rounding off, percentages might not add up to 100%. [b]: shown as percentage of all reoperations. [c]: in complete months. [d]: in millimeters, measured six weeks postoperatively. * Statistically significant difference (*p* < 0.05).

**Table 3.** Description of reoperations.

| Case | CHS | Age | Sex | ASA | Lateral Protrusion [a] | Lateral Protrusion [b] | Type of Reoperation | Reason of Reoperation | Mortality | Time to Reoperation [c] |
|------|-----|-----|-----|-----|------------------------|------------------------|---------------------|-----------------------|-----------|-------------------------|
| 1 | 2 | 82 | F | 3 | 8 mm | 8 mm | THP | Coxarthrosis | No | 25 |
| 2 | 2 | 81 | F | 2 | 6 mm | 6 mm | HA | Screw cut out | No | 1 |
| 3 | 2 | 76 | F | 3 | 2 mm | 5 mm | Screw replacement | Pain complaints | No | 4 |
| 4 | 2 | 78 | F | 3 | 2 mm | 6 mm | THP | Coxarthrosis | No | 5 |
| 5 | 2 | 40 | M | 2 | 6 mm | 12 mm | Screw replacement | Pain complaints | No | 6 |
| 6 | 2 | 68 | F | 2 | 5 mm | 5 mm | Screw replacement | Pain complaints | No | 7 |
| 7 | 2 | 70 | F | 2 | 7 mm | 10 mm | Screw replacement | Pain complaints | No | 13 |
| 8 | 2 | 56 | F | 1 | 13 mm | 17 mm | Screw replacement | Pain complaints | No | 3 |
| 9 | 2 | 53 | F | 1 | 0 mm | 0 mm | Screw replacement | Pain complaints | No | 15 |
| 10 | 2 | 71 | F | 3 | 4 mm | 5 mm | HA | Fall | Yes | 2 |
| 11 | 2 | 82 | M | 3 | 10 mm | 15 mm | Screw replacement | Pain complaints | No | 5 |
| 12 | 2 | 86 | F | 3 | 1 mm | 11 mm | HA | AVN | No | 2 |
| 13 | 2 | 89 | M | 3 | 5 mm | 9 mm | HA | Screw dislocation | No | 1 |
| 14 | 2 | 87 | M | 4 | 13 mm | 15 mm | HA | Screw cut out | Yes | 1 |
| 15 | 3 | 62 | F | 2 | 1 mm | 1 mm | Screw replacement | Pain complaints | No | 7 |
| 16 | 3 | 66 | F | 4 | 1 mm | 6 mm | Screw replacement | Pain complaints | No | 8 |
| 17 | 3 | 78 | F | 1 | 1 mm | 7 mm | Screw replacement | Pain complaints | No | 35 |
| 18 | 3 | 78 | F | 3 | 8 mm | 10 mm | Screw replacement | Pain complaints | No | 7 |
| 19 | 3 | 74 | F | 2 | 0 mm | 0 mm | Screw replacement | Pain complaints | No | 11 |
| 20 | 3 | 89 | F | 3 | 6 mm | 12 mm | THP | Screw cut out | No | 4 |
| 21 | 3 | 66 | F | 2 | 4 mm | 5 mm | THP | Coxarthrosis | No | 32 |
| 22 | 3 | 44 | F | 1 | 16 mm | 19 mm | Screw replacement | Pain complaints | No | 10 |
| 23 | 3 | 69 | F | 1 | 0 mm | 4 mm | Screw replacement | Infection | No | 1 |
| 24 | 3 | 73 | F | 2 | 10 mm | 13 mm | THP | Pain complaints | No | 5 |
| 25 | 3 | 75 | F | 2 | 8 mm | 12 mm | Screw replacement | Pain complaints | No | 1 |
| 26 | 3 | 76 | F | 2 | 0 mm | 4 mm | THP | Coxarthrosis | No | 12 |
| 27 | 3 | 59 | F | 2 | 0 mm | 0 mm | THP | Nonunion | No | 6 |

Abbreviations: CHS, cannulated hip screw; ASA, American Society of Anesthesiologists; AVN, avascular necrosis of the femoral head. [a]: measured six weeks postoperatively. [b]: measured before reoperation. [c]: in complete months.

In the total cohort, 33 patients were older than 80 years. Of them, seven (21%) underwent reoperation compared to 20 patients (30%) below the age of 80. This difference was not significant ($p = 0.499$). The type of reoperation differed significantly between these groups ($p = 0.005$). Of patients above the age of 80 years who underwent reoperation, conversion to HA was performed in four subjects (57%), screw replacement in one subject (14%), and conversion to THP in two subjects (29%). For patients under the age of 80, one subject (5%) underwent conversion to HA, 14 subjects (70%) had screw replacement, and five subjects (25%) underwent conversion to THP.

The lateral protrusion of the screws outside the lateral cortex was 2 mm for the two CHS group and 1 mm for the three CHS group six weeks postoperatively ($p = 0.330$). The mean follow-up duration differed significantly, being 26 months for the two CHS group and 45 months for the three CHS group ($p < 0.001$, Table 2).

## 4. Discussion

The main objective of this study was to determine whether the fixation of non-dislocated femoral neck fractures (Garden I and II) by two CHS leads to similar results as fixation by three CHS. The most important finding of our study was the number of reoperations after two versus three CHS. We did not find a clinically significant difference between these two surgical techniques (28% versus 26%, $p = 1.000$). In terms of reoperation, infection, and screw dislocation, there were no differences in outcome between fixation by two or by three CHS. Also, no higher reoperation rate was seen in patients older than 80 years of age.

This is the first prospectively maintained database that investigated reoperation rates between non-displaced femoral neck fractures for two CHS versus three CHS. One randomized controlled trial has been performed comparing two versus three CHS. This study reported a significantly higher frequency of radiological union in two CHS fixation [27]. This study, however, did not mention the clinical outcomes or reoperation rates. One retrospective study did not find a significant difference in radiological outcome between two and three CHS [15]. In two studies, the clinical outcome was investigated for two CHS fixation. Both studies reported low rates of complications, indicating that two CHS suffice

in the fixation of non-displaced femoral neck fractures [23,26]. Yet, these studies had no control group.

Biomechanical studies have examined the maximum force that both types of fixation can resist. In cadaveric or synthetic bones, femoral neck fractures were created and fixated with two or three CHS. Three CHS fixation showed greater resistance to mechanical loading in all studies [19–21]. This result is not surprising. The value of this ability to bear more load, however, seems insignificant for clinical practice, as seen in the present study.

Although no significant clinical difference was observed between these two groups in our study, our overall rate of reoperations was found to be higher compared to the literature. A recent systematic review by Kim et al. focused on internal fixation with cannulated screws for non-displaced femoral neck fractures in patients over 60 years of age [29]. They reported a reoperation rate of 15%, with 12% of cases requiring conversion to arthroplasty. Unfortunately, the review did not specify whether fixation was performed using two or three CHS. Studies specifically investigating fixation by two CHS or comparing two versus three CHS are limited. The available literature indicates reoperation rates ranging from 5% to 10% [23,26,30].

In the present study, the overall reoperation rate was notably higher at 27%. The majority of reoperations were due to the number of screw replacements (15 out of 27 reoperations). It would be worthwhile to explore whether the utilization of hydroxyapatite-coated screws could help reduce this high reoperation rate. Previous research on lateral femoral fractures has shown improved osteointegration with hydroxyapatite-coated screws compared to conventional screws [31]. This might decrease screw loosening and, thus, the need for screw replacement. Therefore, investigating the potential benefits of hydroxyapatite-coated screws in femoral neck fractures could be an interesting avenue for future research. When screw replacement was not counted as reoperation, our reoperation rates compare to previously described literature, as conversion to arthroplasty was performed in 12% of cases.

Complications after CHS fixation include non-union and AVN. In our study both non-union and AVN occurred once (1% for the total cohort). This is lower than the rate reported in literature. Non-union rates after fixation with two CHS vary between 0 and 22%, and AVN occurs in between 0 and 6% of cases [23,24,26,30].

Lateral protrusion for the total cohort showed a limited amount of screw dislocation, being 1 mm. It seems contradictory that the majority of reoperations in the current study were screw replacements due to pain of the OSM, since dislocation was minimal. Therefore, dislocation was measured again before reoperation. Some patients showed a progression in dislocation over time. This suggests that, although radiological union was confirmed for all but one of the patients, fractures may not be completely healed after six weeks. This might indicate the need for another visit to the outpatient clinic at six months postoperatively.

The three CHS group had a significantly longer follow-up duration (45 months compared to 26 months for the two CHS group). This was due to the fact that those patients underwent surgery between 2017 and 2019, because all CHS surgeries since 2019 were performed using two CHS. Since most reoperations for non-displaced fractures after internal fixation occur within the first year [32], we chose a follow-up duration of at least a year for all inclusions. Still, the longer follow-up duration for the three CHS group might explain the higher rate of conversion to THP.

This study knows several limitations. First and foremost, the study was not a randomized controlled trial. To minimalize the risk of bias, no patient selection took place, and all patients were included. Second, follow-up duration was counted to the date of data extraction through the electronic patient files and not to the date of the last visit to the outpatient clinic. Patients were instructed to contact the outpatient clinic in case of an adverse event. However, there is a possibility that patients went to a private clinic, without contacting the outpatient clinic. Consequently, the number of reoperations may have been underestimated. Moreover, the duration of surgery was not registered in this study. The expected advantage of reducing surgery time, therefore, cannot be confirmed by

this study. This was a conscious decision since our hospital is a teaching hospital. Surgeries are partially performed by residents, inevitably resulting in prolonged surgery time and, therefore, skewed results.

## 5. Conclusions

In the current study, an evaluation of a prospectively maintained database was performed to compare clinical outcomes of non-displaced femoral neck fractures fixated with either two or three CHS. Our findings revealed that patients who underwent fixation with two CHS had comparable reoperation rates to those who underwent fixation with three CHS. These results suggest that the clinical outcomes achieved with both fixation methods were similar. However, there are advantages of two CHS fixation, although we did not investigate them in the current study. These advantages include reduced operation time, decreased irradiation, and a less demanding technique. Therefore, it can be concluded that two CHS fixation may be the preferable method. This study highlights that the choice for both two and three CHS is justified. Yet, there are potential benefits of utilizing two CHS for the treatment of non-displaced femoral neck fractures, offering a promising alternative to the traditional three CHS fixation approach.

**Author Contributions:** Conceptualization, W.K., G.v.O. and L.H.; methodology, H.S., W.K., G.v.O. and L.H.; software, H.S.; validation, H.S.; formal analysis, H.S.; investigation, H.S. and W.K.; resources, G.v.O., P.v.K., T.T. and W.K.; data curation, H.S., W.K., G.v.O., P.v.K., T.T. and L.H.; writing—original draft preparation, H.S.; writing—review and editing, W.K. and L.H.; visualization, H.S.; supervision, W.K., G.v.O., P.v.K. and T.T.; project administration, H.S. All authors have read and agreed to the published version of the manuscript.

**Funding:** This research received no external funding.

**Institutional Review Board Statement:** The study was conducted in accordance with the Declaration of Helsinki and approved by the Ethics Committee of Meander Medical Center.

**Informed Consent Statement:** Informed consent was obtained from all subjects.

**Data Availability Statement:** Data is contained within the article.

**Acknowledgments:** We wish to thank Matthijs de Haan for providing us with the figures.

**Conflicts of Interest:** The authors declare no conflict of interest.

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
