# Peer review of "Less Is More for Non-Dislocated Femoral Neck Fractures: Similar Results for Two versus Three Cannulated Hip Screws"

_2673-4095, doi:10.3390/surgeries4040048_

Round 1
Reviewer 1 Report (Previous Reviewer 1)
The authors wrote an interesting manuscript, and worked hard on the revision. Its a nice study, although the content of this scientific work is not new. Its always a discussion using two or three screws. But again, it is not understandable for me, why so many reoperations had to be performed, due to loosing of the screws or pain level. The thoughts, using another material for the screws, possibly could influence the rate of reoperations positively.
beside that, I have noting to criticize.
Author Response
We thank the reviewer for their comments on the manuscript. Indeed, using a different material would be very interesting for future research. No further changes were applied following this feedback.
Reviewer 2 Report (New Reviewer)
I have read an article by Hilde Schutte entitled “Less is more for non-dislocated femoral neck fractures: similar 2 results for two versus three cannulated hip screws”. Below are my comments:
MINOR COMMENTS
- why are there parts of the main text in red?
- Are Figures 1 and 2 your own creations?
- please rewrite the references according to the publisher recommendations
- lines 36-38 – If hypertension increases the risk of developing femoral neck fractures, does not mean that the individuals with hypertension are more susceptible? How makes this explanation of the relationship between hypertension and femoral nick fracture bidirectional?
- lines 62-63 – “In cases where the femoral neck could not accommodate three CHS due to the diameter of the femoral neck” – please rewrite and try to avoid repeating words
- line 152-153 – “the majority was female” – please rewrite
- line 153 – please explain ASA
- Figure 4b, section C – are you sure that here are 3 CHS?
MAJOR COMMENTS
- please provide in the introduction section the secondary outcomes of your study
- please provide the number of the ethical committee approval, in the study design section or at the end of the manuscript
- please provide the exact period of the study 9e.g. from 1st of January 2019 to….)
- how many grams of Cefazolin were given as prophylaxis?
- when was the physiotherapy started postoperatively?
- there were a lot more patients with ASA III in the 2 CHS group than in the 3 CHS group. Are you sure that there is not a significant difference? - please provide de P value for every ASA score
- please provide P value for every KDIGO classification
- a more detailed analysis of the patients which were in need for reoperation can be performed in order to provide a clear image of the main risk factors when using 2 CHS
- discussion section should be rewritten and a more detailed comparative analysis with previous published studies should be performed. In this form, it is in part a reinterpretation of the results and decrease the value of the manuscript.
- the advantages of using 2 CHS are not quite a result of your study, mainly reduced operation time, decreased irradiation, and a less demanding technique, because you did not analyze these aspects, didn’t you? Therefore, I consider that in the conclusion section you should focus on your main results.
Minor editing of English language required
Author Response
We thank the reviewer for their thoughtful comments. We tried to follow their suggestions as good as possible. Modified text is highlighted in red.
Reviewer comments
MINOR COMMENTS
- Why are there parts of the main text in red?
Response to reviewer:
This manuscript has undergone revision, but the previous reviewers did not respond. Therefore, new reviewers were assigned. The parts in red are modified parts after the first revision was done.
- Are Figures 1 and 2 your own creations?
Response to reviewer:
Figures 1 and 2 are created specifically for this manuscript by Matthijs de Haan (see acknowledgements).
- please rewrite the references according to the publisher recommendations
Response to reviewer:
References are altered according to publisher recommendations found in the instructions for authors on https://www.mdpi.com/journal/surgeries/instructions
- lines 36-38 – If hypertension increases the risk of developing femoral neck fractures, does not mean that the individuals with hypertension are more susceptible? How makes this explanation of the relationship between hypertension and femoral nick fracture bidirectional?
Response to reviewer:
This suggestion was done by a previous reviewer. We understand the confusion, and therefore left out the sentence. Hypertension is not the focus of the current manuscript.
- lines 62-63 – “In cases where the femoral neck could not accommodate three CHS due to the diameter of the femoral neck” – please rewrite and try to avoid repeating words
Response to reviewer:
Altered to: “In cases where the femoral neck could not accommodate three CHS due to its diameter, the insertion of two CHS has been justified”
- line 152-153 – “the majority was female” – please rewrite
Response to reviewer:
We are not sure what the problem is with the current sentence, but guessed it was not clear enough. New sentence: “The majority of the sample (74%) was female”
- line 153 – please explain ASA
Response to reviewer:
Added: “Distribution over American Society of Anaesthesiologists (ASA) classes was as follows.”
- Figure 4b, section C – are you sure that here are 3 CHS?
Response to reviewer:
Yes, these radiographs are from the same patient. The two lower screws are projected over each other and therefore you might not see them very well.
MAJOR COMMENTS
- please provide in the introduction section the secondary outcomes of your study
Response to reviewer:
At the end of the introduction (line 71), we added: “Screw displacement was considered as a secondary outcome parameter.”
- please provide the number of the ethical committee approval, in the study design section or at the end of the manuscript
Response to reviewer:
The number of the ethical committee approval (TWO 20-063) is added in the methods in line 76.
- Please provide the exact period of the study 9e.g. from 1st of January 2019 to….)
Response to reviewer:
Added in lines 76-77: “The duration of the study extended from January 1st, 2019 through December 26th, 2020.”
- How many grams of Cefazolin were given as prophylaxis?
Response to reviewer:
Amount of Cefazolin added in line 88
- When was the physiotherapy started postoperatively?
Response to reviewer:
Assed to line 102-103: “Directly postoperatively, a physiotherapist is consulted to support patients in mobilizing.”
- there were a lot more patients with ASA III in the 2 CHS group than in the 3 CHS group. Are you sure that there is not a significant difference? - please provide de P value for every ASA score
Response to reviewer:
The Mann-Whitney U test was used to compare ASA en Kidney function class between both groups. This is the appropriate method of comparing two ordinal data groups. For the total comparison, one P value is calculated.
- please provide P value for every KDIGO classification
Response to reviewer:
See response to comment #14
- a more detailed analysis of the patients which were in need for reoperation can be performed in order to provide a clear image of the main risk factors when using 2 CHS
Response to reviewer:
Information on reoperations is provided in Table 3.
- discussion section should be rewritten and a more detailed comparative analysis with previous published studies should be performed. In this form, it is in part a reinterpretation of the results and decrease the value of the manuscript.
Response to reviewer:
The discussion was altered and comparison with previous studies is added in lines 220-258.
- the advantages of using 2 CHS are not quite a result of your study, mainly reduced operation time, decreased irradiation, and a less demanding technique, because you did not analyze these aspects, didn’t you? Therefore, I consider that in the conclusion section you should focus on your main results.
Response to reviewer:
We altered the conclusion, focusing more on the results of the study, and making it clear that the aforementioned advantages are not a result of the current study (lines 294-298).
Reviewer 3 Report (New Reviewer)
Dear colleagues!
The biomechanics of injury and rehabilitation are a hot topic of our time. Therefore, your research is especially interesting.
The manner of presenting the manuscript is academic, the language is scientific and does not require editorial work. But if from a formal point of view everything is fine, then upon closer examination, some points that raise questions are striking.
1. Formatting the article
Why do you have some paragraphs highlighted in red?
2. List of references
Today is 2023 and we are asking the relevance of the study along the tracks of engineering and healthcare development over the past few years. But you have links to articles (1, 3, 6, 7, 8, 11, 13, 20, 21, 23, 25, 28) that are more than 10 years old, and in some cases were published in the last century .
Given this state, it is difficult for me to fully appreciate the relevance of the study, as well as the discussion.
You need to update the list of references and rewrite related sections with this in mind
3. Introduction
Why don't you specify the null hypothesis?
4. Materials and methods
Why is there no information about the number of participants in the study and a formula for calculating the sample size?
What were the inclusion, non-inclusion and exclusion criteria?
Describe in more detail the procedure for quality control of the operation.
Did you have ethics approval for the study?
5. Results
You write about patients who had pain syndrome. However, you do not evaluate the quality of life of patients even after the intervention. Describe how the operation affected the lives of patients.
Author Response
Author’s note to reviewer report - Reviewer 3
We thank the reviewer for their thoughtful comments. We tried to follow their suggestions as good as possible. Modified text is highlighted in red.
Reviewer comments
The biomechanics of injury and rehabilitation are a hot topic of our time. Therefore, your research is especially interesting.
The manner of presenting the manuscript is academic, the language is scientific and does not require editorial work. But if from a formal point of view everything is fine, then upon closer examination, some points that raise questions are striking.
- Why do you have some paragraphs highlighted in red?
Response to reviewer:
This manuscript has undergone revision, but the previous reviewers did not respond. Therefore, new reviewers were assigned. The parts in red are modified parts after the first revision was done.
- List of references
Today is 2023 and we are asking the relevance of the study along the tracks of engineering and healthcare development over the past few years. But you have links to articles (1, 6, 7, 8, 11, 13, 20, 21, 23, 25, 28) that are more than 10 years old, and in some cases were published in the last century .
Given this state, it is difficult for me to fully appreciate the relevance of the study, as well as the discussion.
You need to update the list of references and rewrite related sections with this in mind
Response to reviewer:
We reviewed the references.
For reference 1 it is mentioned that this study dates from the last century. Also, the information is still relevant.
References 6 and 7 were suggested by previous reviewers.
Reference 8 includes different treatment options, which are still relevant
Reference 11 refers to the Garden classification, which is still relevant
Reference 13 shows that previously it was common to insert 3 CHS, as this is an old method, it is logical that this reference is dated >10 years
References 20 and 21 are biomechanical studies, which information is still relevant
Reference 23 and 25 indicate that 2 CHS fixation might be adequate. We tried to find more recent publications on this topic, but weren’t able to find them.
Reference 28 includes the KDIGO kidney function classification, and is therefore dated >10 years.
- Introduction
Why don't you specify the null hypothesis?
Response to reviewer:
Null hypothesis added to introduction (lines 71-74):
“The null-hypothesis of the study is stated as follows: “There is no significant difference in clinical outcomes, including reoperation rates and screw dislocation, between patients treated with two CHS and those treated with three CHS for non-dislocated femoral neck fractures.”
- Materials and methods
Why is there no information about the number of participants in the study and a formula for calculating the sample size?
Response to reviewer:
Sample size calculation was not performed. Although the data from 2CHS patients were collected prospectively, this is a retrospective cohort studies. When the surgical procedure switched from 3 to 2 CHS fixation, we started data collection. An estimated sample size would have been about 385. Yet, with 25 CHS fixations per year, it would have taken too long to review the outcomes. We did not want to wait too long to evaluate whether there would be more complications. Therefore, after two years of data collection, outcomes were evaluated to see whether changing to 2 CHS fixation was the right decision.
What were the inclusion, non-inclusion and exclusion criteria?
Response to reviewer:
No selection took place, yet, criteria under which patients enrolled the study are added in lines 79-84:
“Instead of inserting three cannulated screws, Garden 1 or 2 femoral neck fractures were fixated with two screws without patient selection. All patients entered a prospectively maintained database to carefully evaluate this change in practice. All patients who underwent fixation by two CHS up until December 2020 were included in this study. Controls with three CHS were identified in consecutive order until a corresponding amount of patients was obtained.”
Describe in more detail the procedure for quality control of the operation.
Response to reviewer:
The procedure to control positioning of the screws is added in lines 94-100:
“Kirschner (K-)wires are placed under X-ray guidance for optimal positioning of the screws in two directions. The first K-wire is placed inferiorly, over the calcar, right in the middle of the femoral cortex. The second K-wire is placed superior to the first and more posterior. The length of the K-wires is measured to determine screw lengths. The cannulated 6.5 mm screws are placed over the K-wires, starting with the posterior screw to prevent varus deformity. After placement of both screws, positioning is checked by X-ray in both directions.”
Did you have ethics approval for the study?
Response to reviewer:
The number of the ethical committee approval (TWO 20-063) is added in the methods in line 76.
- Results
You write about patients who had pain syndrome. However, you do not evaluate the quality of life of patients even after the intervention. Describe how the operation affected the lives of patients.
Response to reviewer:
We did not investigated patient satisfaction in a standardized manner (e.g. through questionnaires). Therefore, we unfortunately cannot objectively measure patient satisfaction after surgery.
Round 2
Reviewer 3 Report (New Reviewer)
Dear colleagues!
Thank you for the opportunity to review the study again. Your answers satisfy me
This manuscript is a resubmission of an earlier submission. The following is a list of the peer review reports and author responses from that submission.
Round 1
Reviewer 1 Report
The authors of this study have tried to answer an interesting and thematically important question. It was about the surgical treatment of non-dislocated femoral neck fractures, which after 2019 was always done with 2 screws. In this sense, a prospective non-randomized study was conducted, comparing the patient group with a matched patient group from the period before 2019, namely with the patients whose fractures were treated with osteosynthesis with 3 screws.
These were exclusively Garden type I and II fractures. After the patients had been excluded based on the inclusion and exclusion criteria, 50 patients were included in each of the two groups. The main target parameter was the rate of reoperations, secondary target parameters were screw dislocations. The most striking results in both groups were - high reoperation rates - group (2 screws): 14 patients (28%) vs. control group (3 screws): 13 patients (26%). The authors thus report a much higher reoperation rate compared to the literature, which is about 4-6 times higher in this study.
TITLE
General: I would change the title in
“The surgical treatment of non-dislocated femoral neck fractures shows almost identical results - regardless of whether 2 or 3 screws are used”
ABSTRACT
General: Good overview of the paper
INTRODUCTION
General:
The structure of the introduction did not look very well and this part is very short. Basics are missing for the general treatment of femoral neck fractures and their results. It should be briefly explained what the criteria for surgical treatment and which kind of options do surgeons have. No studies with clinical or radiological results are mentioned. At the end of the introduction, a hypothesis of the study should be mentioned.
MATERIALS AND METHODS
There is nothing to criticize about the Material and Methods part, this part contains the important information and is well written.
RESULTS:
General:
There is nothing to criticize about the results part, this part contains the important information and is well written.
As already mentioned in the introduction, both groups show almost similar results - especially with regard to the reoperation rate, nevertheless the complication rates are very high compared to the literature.
DISCUSSION:
General: Some good points were mentioned, especially according treatment strategies, however I feel like that very important facts are missing. Actually, it has to be said that this part is not really a scientific discussion. There is a lack of comparative work from the literature, whereby the results should be compared with the results of this study. In addition, the clinical relevance should be worked out more.
The discussion part should start with: .. The most important finding of this study was ….
CONCLUSION
General: is missing
REFERENCES
General: Too few literature references were made and used. Check for Updates
TABLES
General: Tables are okay.
FIGURES
General: Figures are okay
I would also highly recommend some native English proofreading to help with clarity
Author Response
We would first like to thank the reviewer for their valuable time in reviewing our manuscript. We have used the comments to improve our manuscript. We provided a list of all comments and how we’ve dealt with them. Major changes in the manuscript are colored red. Smaller English language textual changes are not colored.
Author’s note to reviewer report - Reviewer 1
Reviewer comments
- I would change the title in
“The surgical treatment of non-dislocated femoral neck fractures shows almost identical results - regardless of whether 2 or 3 screws are used”
Response to reviewer:
We changed the title, using your suggestions.
The new title is: “Less is more for non-dislocated femoral neck fractures: similar results for two versus three cannulated hip screws”.
- The structure of the introduction did not look very well and this part is very short. Basics are missing for the general treatment of femoral neck fractures and their results. It should be briefly explained what the criteria for surgical treatment and which kind of options do surgeons have. No studies with clinical or radiological results are mentioned. At the end of the introduction, a hypothesis of the study should be mentioned.
Response to reviewer:
We altered the introduction, giving more information about current guidelines and treatment options, and we added some references to clinical and radiological studies. The hypothesis/main objective of the study is mentioned in the last sentence of the introduction.
- The discussion part should start with: .. The most important finding of this study was ….
Response to reviewer:
Added to discussion.
- Conclusionis missing
Response to reviewer:
We added a conclusion.
- References: Too few literature references were made and used. Check for Updates
Response to reviewer:
We added literature references in order to give a better scientific background for the study.
- I would also highly recommend some native English proofreading to help with clarity
Response to reviewer:
Textual changes made by a native English speaker.
Reviewer 2 Report
In this comparative study, the authors compared the efficacy of two cannulated hip screws (CHSs) compared with three CHSs for management of non-displaced femoral neck fractures (FNFs). Outcomes of 100 patients demonstrated that fixation by two CHSs is an acceptable treatment method for non-dislocated femoral neck fractures and the insertion of a third screw does not lead to superior clinical results. Although the outcomes are encouraging, I would not recommend using two CHSs for treatment of non-displaced FNFs. The primary reason is the anti-rotation of two CHSs are weaker than that of three, which increase the risk of adverse events.
Detailed comments:
Actually, in clinics, the number of patients with Garden I FNF is seldom. Therefore, the majority of the patients included were Garden II FNF.
For figure 4, the authors should provide the preoperative X-rays and follow-up x-rays.
The language is acceptable.
Author Response
We would first like to thank all three reviewers for their valuable time in reviewing our manuscript. We have used the comments to improve our manuscript. We provided a list of all comments and how we’ve dealt with them. Major changes in the manuscript are colored red. Smaller English language textual changes are not colored.
Author’s note to reviewer report - Reviewer 2
Reviewer comments
- Although the outcomes are encouraging, I would not recommend using two CHSs for treatment of non-displaced FNFs. The primary reason is the anti-rotation of two CHSs are weaker than that of three, which increase the risk of adverse events.
Response to reviewer:
The traditional method with inserting three screws has been done because of the anti-rotation forces. When tested in mechanical studies, anti-rotation forces are indeed stronger when 3 screws are inserted. However, the clinical relevance seems doubtful. Our clinical study does not reveal differences in adverse events or reoperation between 2 or 3 CHS fixation. However, considering the advantages of two CHS fixation, including reduced operation time, decreased irradiation, and a less demanding technique, it can be concluded that two CHS fixation may be the preferable method.
- For figure 4, the authors should provide the preoperative X-rays and follow-up x-rays.
Response to reviewer:
Figures added as requested
Reviewer 3 Report
Dear Authors, the topic is estremely interesting in fact the the use of two CHS rather than three does not seem to change the stability of the synthesis in clinical practice, also reduces surgery time and operative irradiation, reducing exposure for the patient and for the operating room staff.
As regard the introduction I suggest to enrich this section by deepening the epidemiological aspects of femoral fractures, considering the forecast of the increase in their incidence. In this context, for example, you could mention this interesting study in which the authors observed that in hypertensive patients the most frequent site of fracture was the femur:
- - European Journal of Inflammation, Volume 15, Issue 1, Pages 53 – 56. 1 April 2017, DOI 10.1177/1721727X17698473
Frequency of hypertension in hospitalized population with osteoporotic fractures: Epidemiological retrospective analysis of Hospital Discharge Data in the Apulian database for the period 2006-2010, Notarnicola A. et al.
As regards the M&M, when you describe the surgical technique no mention is made of the prophylaxis of venous thromboembolism in both pre and post-operative. I suggest to specify it.
The section results is well described.
As regards the discussion I suggest to improve this section proposing as a possible choice of the osteosynthesis material, the use of screws coated with hydroxyapatite with greater mechanical stability than standard screws, ensuring better osteointegration of the plant. For example, you could report this study in which the authors observed this aspect, albeit for lateral fractures of the femoral neck:
- - Journal of biological regulators and homeostatic agents, Volume 28, Issue 1, Pages 125 – 132, 2014 Jan-Mar
The effect of hydroxyapatite coated screw in the lateral fragility fractures of the femur. A prospective randomized clinical study. Pesce V et al.
Also, there is no section of the conclusions, so I suggest adding it.
Author Response
We would first like to thank all three reviewers for their valuable time in reviewing our manuscript. We have used the comments to improve our manuscript. We provided a list of all comments and how we’ve dealt with them. Major changes in the manuscript are colored red. Smaller English language textual changes are not colored.
Author’s note to reviewer report - Reviewer 3
Reviewer comments
- As regard the introduction I suggest to enrich this section by deepening the epidemiological aspects of femoral fractures, considering the forecast of the increase in their incidence. In this context, for example, you could mention this interesting study in which the authors observed that in hypertensive patients the most frequent site of fracture was the femur:
- - European Journal of Inflammation, Volume 15, Issue 1, Pages 53 – 56. 1 April 2017, DOI 10.1177/1721727X17698473
Frequency of hypertension in hospitalized population with osteoporotic fractures: Epidemiological retrospective analysis of Hospital Discharge Data in the Apulian database for the period 2006-2010, Notarnicola A. et al.
Response to reviewer:
We added the suggested study to the introduction and enriched the introduction.
- As regards the M&M, when you describe the surgical technique no mention is made of the prophylaxis of venous thromboembolism in both pre and post-operative. I suggest to specify it.
Response to reviewer:
Thromboembolism prophylaxis added in M&M
- As regards the discussion I suggest to improve this section proposing as a possible choice of the osteosynthesis material, the use of screws coated with hydroxyapatite with greater mechanical stability than standard screws, ensuring better osteointegration of the plant. For example, you could report this study in which the authors observed this aspect, albeit for lateral fractures of the femoral neck:
- - Journal of biological regulators and homeostatic agents, Volume 28, Issue 1, Pages 125 – 132, 2014 Jan-Mar
The effect of hydroxyapatite coated screw in the lateral fragility fractures of the femur. A prospective randomized clinical study. Pesce V et al.
Response to reviewer:
We mentioned the use of different screws and it’s results on osteointegration, and that this might reduce the number of reoperations due to screw replacement.
- Also, there is no section of the conclusions, so I suggest adding it.
Response to reviewer:
Conclusion section added